# Research on the Development of Urban Parks Based on the Perception of Tourists: A Case Study of Taihu Park in Beijing

**DOI:** 10.3390/ijerph19095287

**Published:** 2022-04-26

**Authors:** Yaqi Du, Rong Zhao

**Affiliations:** Research Institute of Forestry Policy and Information, Chinese Academy of Forestry, Beijing 100091, China; duyq_0806@163.com

**Keywords:** Beijing, urban park, satisfaction, tourist perception

## Abstract

With the development of the economy and society, the derivative needs beyond the basic survival needs of citizens are constantly expanding. The emergence of urban parks caters to the needs of citizens to relax, playing an important role in improving the ecological environment, providing leisure and recreation places, and having a good prospect of development. This paper takes Taihu Park in Beijing as an example, from the perspective of tourists. The influence factors are analyzed with the structural equation model, the influence of factors, and drawn up to a degree. The tourists’ satisfaction and loyalty were positively related to the change; the tourists’ satisfaction and complaints about change had a negative correlation and were put forward to strengthen the construction of infrastructure to park development. It is suggested to improve the functional level of the park and increase the selling point of commodities in the park.

## 1. Introduction

The urban park first appeared in western countries, during the industrialization of western countries, as the symbol of economic development and progress. Since the concept of “Urban Open Space” was first proposed in the UK in 1877, the research and the design of urban parks have been developed successively in other European and American countries [1]. After a long period of development, urban parks in these countries now have a relatively mature construction and management system. Relevant laws and regulations have a good effect under the management of the supervision department. Studies on urban parks mainly focus on people, including human behavior, social value, what helps to develop a sense of community [2,3,4,5,6], how to classify urban parks to provide better services for human beings from the perspective of anthropology and sociology, and the value influence mechanism between the urban park and human behavior [7,8]. In recent years, an innovative landscape has been proposed to attract more park users [9]. Additionally, research shows that elements such as biodiversity, environment, accessibility, usability, esthetic quality, recreational, and ecological functions will resolve park users’ perception of the space [10,11,12,13]. The construction of urban parks has been in full swing in China in recent years [14]. Since 2019, China has been paying more and more attention to ecological construction; actively exploring the new direction of urban ecological construction is also an important part of it. Urban parks provide leisure services for people and play an important role in the economy, environment [15,16,17,18,19], culture, and other aspects.

As the number of urban parks grows, especially in China, the difficulties and problems faced by parks has also shifted from construction to management. How to efficiently manage and maintain the facilities and the environment in the park will become a problem that the built urban parks need to consider and solve in the future. Earlier studies [20,21,22] on the relationship between event quality, destination image, quality of experience, and tourist satisfaction showed a positive relationship between these variables. For instance, a recent study [20] showed that event quality positively influences destination image, perceived value, tourist satisfaction, and loyalty towards a destination [23]. Relevant studies put forward that the urban park is a green space specially planned and constructed within the city for residents’ viewing, rest, health care, and entertainment. However, to find out the deficiencies existing in the park based on the perspective of tourists, to constantly improve the satisfaction of tourists, and to improve the service provided by the urban park are the long-term developments to which attention must be paid. The urban park takes recreation as its main function and also improves the urban ecological quality, beautifies the urban environment, prevents and reduces urban disaster, and regulates the climate [24]. Recently, research on urban parks has gradually changed from the basic initial concept definition, type division, planning, design, etc., [25] to the size and approach of urban parks’ play value and utility improvement [26,27,28]. As the parks develop, consumers, as important users, have become the focus of research. The feedback from tourists can put forward effective suggestions for improving the efficiency of park management.

Although some scholars combine the post-use evaluation of tourists with the development of urban parks, the research significance of urban parks varies from place to place due to different regions, resource endowment, and city positioning. At present, the research on urban parks is mainly carried out from a macro perspective. To analyze the classification and function or the future development of municipal parks, parks with mature development and high visitor volume are the research object. From a micro perspective, the studies lie on the fairness of urban parks, visitor classification, and desired improvement in urban parks based on a specific park which has developed maturely [29,30,31].

For the research method, the American Customer Satisfaction Model (ACSI) is a very mature and widely used model that studies consumer behavior and intentions. The ACSI model is a multi-indicator-supported model consisting of six latent variables. Among the latent variables, customer satisfaction is the core variable. Customer expectation, perceived quality, and perceived value are the cause variables of customer satisfaction, while customer complaint and customer loyalty are the result variables of customer satisfaction [32]. Customer satisfaction is to quantify the difference between customer expectation and real experience to form numerical indicators to scientifically guide enterprise management. Initial research content is related to customer satisfaction and its influencing factors, etc. Then, according to its extension development, tourist satisfaction was regarded as a separate branch of study that builds a tourist satisfaction evaluation model mainly through analyzing the tourist behavior and its results. In the research of urban parks, customer means tourists. Pitzam et al., believed that tourist satisfaction is the result of comparing tourists’ expectations of the tourist destination with the on-site tourism experience. If the on-site tourism experience is higher than the prior expectation, the tourists are satisfied [33]. Beard et al., also emphasized that tourist satisfaction is based on the positive effect of comparing tourist expectations with actual experience [34]. Accessibility measures how much time people spend on the road and it affects the partial tourist satisfaction of the park. There is also a direct relationship between perceived quality and customer satisfaction. Anderson believes that perceived quality is the most important factor affecting tourist satisfaction [35]. Customer loyalty and customer satisfaction are positively correlated. Tourism researchers believe that when studying the relationship between satisfaction and tourist loyalty, tourist loyalty should be understood as the unity of behavioral loyalty and emotional loyalty [36,37]. Complaints arise when customers are dissatisfied. Sean pointed out that when customers are dissatisfied with a product or service, there will be three consequences: voice, exit, and word of mouth, which all reduce the tourist satisfaction [38]. In the study of tourist satisfaction, scholars generally found that customer loyalty and customer complaint have an important influence on customer satisfaction [39]. Previous studies mainly focus on the relationship of the variable factors, including perceived quality, behavioral tendency, tourism pushes, pull motivation, loyalty, and satisfaction in the expectancy disparity model [37,40,41,42,43]. It is concluded that perceived quality, loyalty, and tourism pushes have a positive impact on tourist satisfaction yet negative behavioral tendency reduces tourist satisfaction. Some research discuss the factors in more detail, such as covering the tourist expectations, tourism landscape, supporting facilities, tourism services, and management [44,45]. In addition, the latest studies take the impact degree of transportation convenience, culture, operating environment, and tourist motivation into consideration to analyze the relationship between the factors [46,47].

Based on the abovementioned studies, we use Taihu Park in Beijing to explore whether a causal relationship exists between accessibility, tourist perception, tourist loyalty, tourist complaint, and tourist satisfaction. Compared with previous research, it is meaningful that the study creatively chooses Taihu Park. Based on the urban positioning of Beijing as a “political center”, “cultural center”, and “international communication center”, Tongzhou District, where Taihu Park is located, is planning to build an “urban sub-center”. With the development of urban construction, Taihu Park is transforming into an urban park from its earlier positioning as a country park and will carry more park functions and more tourists. However, Taihu Park also faces the problem of lack of unique resource endowment and insufficient funds for renovation. These characteristics of Taihu Park are not possessed by the park objects of previous studies, which focused on well-funded mature large parks or conducted macro studies of park systems across the city [29,30,31]. Therefore, the conclusions of these studies cannot be directly applied to Taihu Park. In terms of research methods, it is novel and considered applicable to use the American Customer Satisfaction Index (ACSI) model to analyze Taihu Park. 

## 2. Materials and Methods

### 2.1. Study Area

At the lowest point of Beijing, Taihu Town is low lying and famous for its aquaculture industry (Figure 1). Meanwhile, Taihu is also an important transfer center of the Beijing-Tianjin Expressway with convenient transportation, relatively developed local agriculture, and favorable development conditions for secondary and tertiary industries [48]. With the construction of the sub-center of Tongzhou District in Beijing in recent years, Taihu Town has provided a lot of development opportunities (Table 1).

Taihu Park is one of the landmark attractions in Taihu Town (Figure 2). Taihu Park is located in the southwest corner of the intersection of Jiuzhou Road and Zhangtai Road, Tongzhou District, Beijing, covering an area of about 147 hectares (Figure 3). It is one of the large-scale greening projects in the sub-center of Beijing city. Many landscapes are set up in the park, forming twelve characteristic scenic spots including: “Taihu Xiagang”, “Hongqiao Fish Music”, and “Gaotai Qiubo”. Taihu Park has been open to tourists since July 2014. In the early stage, it only provides recreational activities without any paid entertainment items. It is a public welfare rural country park. As the park gradually opened and the number of tourists increased, the campus management and maintenance came under certain pressure. Considering the demand for entertainment, the unit of the management began to provide tourists with a cruise ship and bike rental to choose from, and opened the visitor center, providing reading, a rest area, and creating other products to buy. At present, there are no other income-generating projects and the maintenance and management of trees in the park is still the main work content.

Taihu Park was positioned as a rural country park in the early stage of planning. Later, with the construction of the city sub-center in Tongzhou District, the increase of surrounding residents, and the related needs of development planning, it was transformed into an urban park, which improved the functional level of the park and provided more leisure space. However, Taihu Park still faces problems such as backward infrastructure, low tree density, and short tree age. At the same time, insufficient operation and maintenance funds also limit the development of Taihu Park. How to transform and upgrade Taihu Park with limited funds has become an important research direction for the development of Taihu Park.

### 2.2. Method

The American Customer Satisfaction Index (ACSI), which is recognized worldwide, measures the quality of products and services based on consumers’ consumption experience. The ACSI model is divided into perceived performance, customer expectation, customer complaint, customer loyalty, and overall customer satisfaction. Perceived performance and customer expectation together constitute perceived value and affect customer satisfaction together with customer complaint and customer loyalty [49,50]. Indeed, studying the leisure perception, experience, open-space needs of residents, and so on in a certain area may help provide suggestions for local green space planning and design [51,52], which can help researchers understand common reasons and impact factors and their significance that affect residents’ recreational perception and satisfaction [13]. 

Based on the ACSI model (American Customer Satisfaction Index), this paper analyzes the influencing factors of tourist satisfaction from the four dimensions of Accessibility, Tourist Loyalty, Tourist Perception, and Tourist Complaint (Figure 1).

### 2.3. Research Hypothesis

There is a causal relationship between tourist satisfaction and accessibility, tourist perception, tourist loyalty, and tourist complaint. Tourist satisfaction refers to the satisfaction of the park as a whole and the fulfillment of the expectations, which is influenced by the four indicators. 

Accessibility refers to the ease with which visitors can reach the park. With the development of urban parks, the number of urban parks is increasing and the site selection of parks largely determines the size of the flow of people. Accessibility measures whether the time it takes people to reach the park is acceptable in any travel mode they choose. With more convenient travel, accessibility is an important factor to consider the satisfaction of a park. Therefore, this paper uses accessibility as an important indicator for measuring the tourist satisfaction of Taihu Park.

Tourist perception includes two kinds of perception: quality and service perception and environmental awareness. Quality and service perception refers to the tourist perception of all aspects of landscape quality and service quality after traveling the Taihu Park and is one of the most influential indicators of tourist satisfaction. Environmental awareness refers to the perception of the environmental sanitation facilities, signage, and sanitary conditions, which represent the park’s consideration for tourists.

Tourist loyalty is the persistent behavior that tourists obviously prefer to participate in specific recreational activities. In the study of tourist behavior, tourist loyalty can be divided into two levels: attitude and behavior. The attitude is the emotional preference of tourists. The behavior refers to the number of times tourists participate in specific activities, facilities, and services, which shows the consistency of tourists’ multiple participation. For urban parks, the revisit rate and whether the tourists would like to come back when the park starts collecting tickets, represent the measure of the attitude and behavior. This paper uses “revisit” and “recommend to others” as main indicators for measuring tourist loyalty.

Tourist complaint refers to the adverse consequences that may occur when the tourists fail to achieve the expected goal after traveling the park [53]. In the past, park managers did not pay attention to tourists’ complaints, but with the development of network technology bad reviews will spread rapidly on social networks, which has a negative impact on the development of the park. Therefore, this paper takes tourist complaint as an important indicator, including whether there are suggestions or comments that need to be put forward to the park managers and whether it is worthy of coming back to.

After traveling the park, if the tourists think the Taihu Park provides good service to meet their expectations and the landscape is worthy of revisiting, they will have great comments on the park in the four aspects. The higher the perceived quality the higher the satisfaction they feel. Therefore, this paper makes the following assumptions about the relationships between tourist satisfaction, accessibility, tourist perception, tourist loyalty, and tourist complaint:

**Hypothesis** **1.***accessibility has a significant positive impact on tourist satisfaction;*


**Hypothesis** **2.**
*environmental perception has a significant positive impact on tourist satisfaction;*


**Hypothesis** **3.***tourist loyalty has a significant positive impact on tourist satisfaction;*


**Hypothesis** **4.**
*service and quality perception has a significant positive impact on tourist satisfaction;*


**Hypothesis** **5.**
*tourist complaint has a significant negative impact on tourist satisfaction.*


### 2.4. Data Collection

A questionnaire method was used to collect data. There were three parts in the questionnaire: the first part was about the usage features of park visitors, including travel time, use motivation, travel mode, stay time, and use frequency; the second part enquired the visitors’ overall satisfaction and satisfaction of different sections in the park; the third part collected the social and demographic data of the Taihu Park’s tourists (Table 2). Respondents respond to the questions with a Likert 5-point scale, which was scored from 1 to 5 representing “very unsatisfied” to “very satisfied”. A higher score indicated a higher perception of satisfaction.

We collected a pilot test with 200 park visitors before the formal survey and adjusted the questionnaire based on the collected data. A formal survey was conducted in a face-to-face method by ten university students, after being trained, to ensure the reliability of the survey results. The survey was conducted to ensure all types of visitors were covered at all times, including weekdays and weekends from 7:00 A.M. to 8:00 P.M. for a total of 10 days from 16 April 2021 to 22 April 2021. Respondents were selected in different places, including near the landscape, next to rest facilities, exits, parking lots, and visitor centers of Taihu Park. To ensure the consistency of the samples, the respondents were restricted to tourists who had just finished visiting Taihu Park. Whether residents nearby or tourists far away, they were regarded as tourists as long as they entered the park, because the park was the subject we were researching and the difference of people’s addresses was not a classification standard. We collected 46 questionnaires in each place mentioned above and ensured that half of the respondents were male and half were female. A total of 230 questionnaires were returned, of which 209 questionnaires were identified as valid after discarding incomplete and unqualified questionnaires. According to the data provided by the management personnel of Taihu Park, there were about 6000 visitors in April 2021 with an average of 200 visitors per day. We collected 33 questionnaires every day, with a sampling rate of about 16.9%, which is enough compared with other sampling tests (7.6% and 17.2%) [54].

### 2.5. Data Analysis

The basic personal information and travel characteristics collected from the questionnaire were analyzed descriptively in Table 3. The questionnaire survey was selected on weekdays and weekends and random sampling was conducted at five scattered locations in Taihu Park to ensure the authenticity and randomness of the questionnaire collection. The ratio of males to females is nearly 1:1 and the park visitors are mainly concentrated in 31 to 40 years old and 50 to 60 years old. The group is mainly divided into the elderly group (50 to 60 years old), which mainly focuses on exercise and fitness, and the young group (31 to 40 years old), who travel with children and the elderly. The education level is mainly concentrated in the undergraduate degree level, and the number of trips is concentrated in the third time or above, reflecting the high loyalty of tourists. Private cars accounted for the largest proportion of travel, accounting for 63.2% of the number of surveyed tourists, more than half. The second is walking, accounting for 14.8%, mainly for residents within 1 to 2 km of the park for fitness and walking. In addition to the two groups, there are a small number of people who use electric bikes and bicycles to travel within 2 to 5 km to the park, which is a relatively suitable place to choose.

According to the analysis of the answers to the travel motivations (Table 4), it is found that the tourists’ travel motives are mainly exercise and fitness and outing and entertainment, and 40.7% and 46.4% strongly agree with these two motives, respectively. Nearly half of the tourists agree with the item of emotional adjustment and relaxation. It can be concluded that tourists take fitness, play, and adjusting the pressure and emotion in life as the main travel motivation.

The reliability and validity of each item should be evaluated before theoretical hypothesis verification [55]. To test the validity and rationality of the evaluation system and questionnaire setting and to ensure the reasonable construction of the evaluation system in this paper, reliability analysis of the recovered data is required. The questionnaire responses of tourism motivation, service and quality perception, environment perception, tourist loyalty, and tourist complaint were analyzed. Cronbach’s α [56] value was used to detect its reliability. A total of 209 valid questionnaires were collected in this survey and the reliability test coefficient was 0.706, which was higher than the reliability level of 0.7, indicating that the scale in this questionnaire had good reliability for analysis purposes.

SPSS23.0 was used to test the validity of the data. The more the KMO value tends to 1 the better the validity of the questionnaire structure. The KMO value [57] of the survey data is 0.894, and the spherical test value significance is significant, which is less than 0.01, so factor analysis can be continued.

### 2.6. Confirmatory Factor Analysis

In this paper, a confirmatory factor analysis (CFA) was carried out for each factor in the model and adjusted according to the factor load and correction index. By constructing the structural equation model shown in Figure 2, the CFA results are shown in Table 5.

If CMIN/DF is less than 3, RMSEA is closer to 0, and other indicators are closer to 1, the fitting degree of the model can be closer to explaining the real situation. As can be seen from the above fitting indexes, CMIN/DF is 2.530, which meets the standard of less than 3, and IFI and CFI both meet the standard of greater than 0.9. However, GFI, AGFI, and TLI are all lower than 0.9, TLI is lower than 0.8, and RMSEA is greater than 0.1. These indicators indicate that there is a big gap between the fitting degree of this model and the actual matrix. According to the analysis results of Modification Indices, the relationship between latent variables can be adjusted and calculated. The results follow.

## 3. Results

### 3.1. Overall Test of Structural Equation Model

In this paper, Amos 21.0, a software developed by SPSS (Irvine, CA, USA) for processing structural equation models, was used to construct the tourist satisfaction model. After model modification, all fitting indexes are improved, which can verify that the adjustment direction is correct. RMSEA value is less than 0.08, indicating a good overall fitting degree. The relevant indicators are shown in Table 6. It can be seen that all indicators meet the standard of the recommended value, so the model match is good. The operation results of the structural equation model are shown in Figure 3.

The path coefficient test results of the structural equation model are shown in Table 7. The three paths of the five paths among the five latent variables are significant at the level of *p* < 0.01.

At first, it was assumed that accessibility, environmental perception, service quality perception, and tourist loyalty were positively correlated with tourist satisfaction, and customer complaint was negatively correlated with tourist satisfaction. After verification, it was found that the assumptions were true, as shown in Table 8. Comparing path analysis coefficients and research hypotheses, all five hypotheses are true to degrees of *p* < 0.01 and *p* < 0.05.

According to the standardized estimate value, the effect of each observation variable on the potential variable can be seen. The detailed explanation and analysis of different observation variables and potential variables will be given as follows.

### 3.2. Discussion of Environmental Perception

Four factors influence customer satisfaction in this model. The influence of tourist perception is one of the biggest effects. Tourists include environmental awareness and quality awareness and service; the environmental awareness effect is 0.594 in Table 8, the biggest in the tourists’ effect. It can be seen that the change of environmental perception associated with direction change of customer satisfaction degree is the largest. The weight of environmental perception is the largest, indicating that it is also the primary determinant of whether people choose to go to the park or not. Therefore, the construction and improvement of the environment can greatly affect the number of tourists visiting Taihu Park.

There are many problems to be solved urgently in the development of Taihu Park. Firstly, Taihu Park has been serving urban residents as an urban park, however it was positioned as a rural country park before. It is in the process of transformation from a rural country park to an urban park, whose functional facilities need to be improved. It needs to be transformed from a park with rural recreation and countryside recreation as its main function into an urban park with leisure function, ecological environment adjustment, and recreation function as a whole, which mainly serves urban tourists and nearby residents. Secondly, in the park the infrastructure is old, the supporting facilities are lacking, and the landscape is made up of mostly open-air or semi-open facilities. The forest with large afforestation investment and tent areas are not enough to shelter from the rain. At the same time, the density of the forest is not large enough to achieve the sunshade function. This has a significant impact on the environmental perception of tourists and the satisfaction of tourists is not high. Thirdly, there is a shortage of funds for park management and maintenance. According to the maintenance cost standard of 3.6 CNY/m^2^ stipulated by the Beijing Municipal Forestry and Parks Bureau, the overall maintenance cost allocated to Taihu Park is about 5.5 million yuan, which is far from meeting the current maintenance task. The lack of funds from the government leads to the failure of basic park services. The existing forest management in the park cannot meet the standard and it is difficult to plant more new trees to achieve the forest density expected by tourists. Taihu Park can improve environmental perception by applying for more operation and maintenance funds to upgrade the infrastructure in the park.

### 3.3. Discussion of Service and Quality Perception

In the case of the most significant correlation between tourist perception and customer satisfaction, the impact effect of service and quality perception is 0.237, which shows that it changes in the same direction as the overall customer satisfaction.

From the survey of tourist perceptions, we know that the services provided by the park are far from meeting the current demand and there is a big gap in tourist expectations. Firstly, there is only one place in the park that provides commodities for sale—self-service vending machines that have been built but have not been put into use in other areas in the park—which causes a lot of inconvenience for tourists who want to buy food and drinks during the journey. Secondly, there are few workers to patrol the environment and offer help to tourists in the park, especially in places where there is no entertainment. According to the conclusion that the perception of service and quality has a significantly positive effect on satisfaction of tourists, it can be concluded that the more the service and quality are developed to meet the needs of consumers the more the satisfaction of tourists will be improved.

### 3.4. Discussion of Accessibility

The latent variable of accessibility includes three observation variables, namely: distribution of entrances and exits, travel time, and travel distance. The changes in accessibility and customer satisfaction are also in the same direction and the impact effect is 0.134. Accessibility to parks reflects whether the park matches the demand of nearby residents [59,60] and how likely distant tourists will come here; the higher the accessibility the higher the customer satisfaction. However, at present the parking lot of Taihu Park is relatively small, the number of parking spaces cannot meet the needs of park tourists, and the parking time increases the travel cost. If the time cost of travel can be reduced, people’s satisfaction can be improved. For example, the park can provide more parking spaces for tourists near the entrance of the park. Taihu Park can focus on providing tourists with some unique software and hardware services to enhance tourist loyalty.

### 3.5. Discussion of Tourist Loyalty

The change of tourist loyalty is the same as tourist satisfaction and its effect on customer satisfaction is 0.124, indicating the higher the customer loyalty the higher the customer satisfaction. Among the observed variables, “willing to revisit here” has the most significant impact on tourist loyalty, which is the closest observation variable reflecting tourist loyalty. Meanwhile, if they are still willing to come here when the park starts charging, for this group it means that the park has at least reached a relatively satisfactory level and the customer loyalty is very high, resulting in a high satisfaction degree of tourists.

### 3.6. Discussion of Tourist Complaint

The change of tourist complaint and tourist satisfaction is reversed and the influence effect is −0.133, meaning when customer complaint increases tourist satisfaction will decrease. In real life, when tourists have a poor perception of the park and are not satisfied with the park environment or their needs cannot be met, they are prone to complain. The observed variables of tourists’ complaints are “whether they think there are many areas that need to be improved in the park” and “whether they will not come here again and do not recommend others to travel”. When people are not satisfied with a place, these two factors will exist at the same time and have a negative impact on tourist satisfaction. In the process of research, the tourists generally feel that the parking lot capacity is too small, the toilet setting is insufficient, the landscape is not beautiful, and it is inconvenient to find the staff when they need to ask for directions. The dissatisfaction is concentrated in the infrastructure and management of the park, which is also the key direction for the development of Taihu Park to improve.

## 4. Discussion

Starting from specific parks, this study studies the places that should be promoted and improved in the development of urban parks. Aiming at Taihu Park as a specific object of study through a questionnaire survey of tourists, it obtains the perception and overall satisfaction of tourists in all aspects of the park and puts forward suggestions for the park. Based on the empirical study of tourist satisfaction, this paper clarifies the relationship between tourist satisfaction and the latent variables. The results support Hypothesis 1 to Hypothesis 5: accessibility, environment perception, service and quality perception, and tourist loyalty have positive impacts on tourist satisfaction, and tourist complaint has a negative impact on tourist satisfaction. In addition, through the path coefficients between latent variables and observed variables (Table 9), sanitary conditions have the greatest impact on environment perception, travel time has the greatest impact on accessibility, and tourist loyalty is mostly affected by the revisit rate. Additionally, service quality has the biggest impact on service and quality perception and negative publicity has the greatest impact on tourist complaint. Moreover, the promotion of the management of the park can help it maximize the utilization efficiency with limited funds given by the government.

Compared with other research, this paper has many innovations. Some studies focus on the impact of urban parks on specific populations such as adolescent and older populations [61,62]. From a macro perspective, this paper analyzes the audience of Taihu Park as a whole in order to put forward overall suggestions for the development of Taihu Park. Compared with other parks, Taihu Park, the object of this study, also has special characteristics such as the mismatch between early planning and current use, which makes Taihu Park require special maintenance and development methods. In previous studies, there were many discussions on the functions of forest parks or suburban parks. For such research objects, the maximization of ecological benefits must be the primary goal of the park, which plays a vital role in the ecological protection of cities and even countries. They tend to have poor accessibility, less tourist traffic than city parks, and inadequate infrastructure. However, for an urban park, serving tourists is its most important goal, and its social and economic benefits are more important than ecological benefits. The special positioning of Taihu Park is the biggest difference between the study and others. This paper is also innovative in research methods. On the basis of the ACSI model, some optimizations are made for the research object of parks: a new indicator for tourists is selected for the consideration of accessibility, a new observation is selected for tourist perception variables, and a tourist satisfaction evaluation system is established.

There are still some shortcomings in this paper that require directions for further research. First of all, this research is not an official research based on government projects, so there is a lack of official data and it is impossible to excavate more about the development status and problems of Taihu Park. The research method of this study also needs to be summarized into a universal research method, so that it can be better generalized to the analysis and research of other parks.

## 5. Conclusions and Suggestions

Taking Taihu Park as an example, this paper analyzes how tourists’ perceptions affect the overall satisfaction of tourists through the data collected by questionnaires and the method of structural equation model. It arrived at the following conclusions:(1)Tourist satisfaction plays an important decisive role in the sustainable development of the park. In order to improve the urban status of the park, it is necessary to improve tourist satisfaction.(2)Environmental perception, service and quality perception, accessibility, and tourist loyalty all have a significantly positive impact on tourist satisfaction. Therefore, to improve the tourist satisfaction, environmental perception, service and quality perception, accessibility, and tourist loyalty must be enhanced.(3)Tourist complaint is in the opposite direction as the change of tourist satisfaction, whose rise will reduce tourist satisfaction.

According to the research results of the tourist satisfaction and its influencing factors, the following suggestions are made to enhance the quality of Taihu Park:(1)At the park manager level, Taihu Park can add self-service vending machines in the park to improve service and quality perception, increase the accessibility by increasing the number of parking spaces, and improve its software and hardware service quality to reduce tourists’ complaints.(2)From the government’s perspective, it is still up to the government to make efforts to enhance the status of urban parks. Therefore, the government should rely on its own influence to increase publicity for Taihu Park, expand its influence, attract foreign investment, and form a development model in which the market promotes the park. In addition, the government should adopt the necessary financial means to encourage Taihu Park to maintain the trees and infrastructures greatly and to improve the tourist satisfaction by giving more financial support and improving the standard of management subsidy.

Urban parks have two essential functions: biodiversity conservation and recreation provision; both have attracted increasing attention in spatial allocation of urban green space and city planning. Biodiversity conservation of urban parks may significantly beautify the environment and improve the recreation experience and health well-being of urban residents [63], while resident recreation behavior will impose a certain disturbance on the protection of urban green habitats [64]. A large number of studies have measured the function of urban parks through selecting the ecological connectivity and spatial accessibility, which provide a scientific basis for formulating multi-functional protection strategies for urban parks. They also try to identify tradeoff relationship of urban parks in terms of ecological and social functions in the Patero ranking method [65,66,67]. For Taihu Park, it is mainly positioned as an urban park to meet the leisure and exercise needs of nearby residents because of considering spatial accessibility, but it also maintains ecological function to beautify the environment to provide better experience. Biodiversity conservation and recreation provision can often exhibit synergies. The improvement of ecological diversity means that the biological system is more and more rich and complex and includes more tree species and animals that provide more choices and attractions for leisure and entertainment, i.e., tree species diversity provides landscape viewing and diversity of animals provides the experience of a tour. Organizing recreational activities such as forest picking, planting experiences, and volunteering can also help promote biodiversity. There is also a trade-off relationship between biodiversity conservation and recreation provision; making appropriate trade-offs can make Taihu Park undertake park functions better. It should give up part of the sightseeing space in order to increase biodiversity, such as reducing the number of boats that can be rented to tourists on the lake in order to protect the ecological diversity of the lake. However, it should abandon some ecological functions, such as reducing lawn area to provide tent building area, to ensure the carrying capacity of tourists for its position. It has enough lawn area to provide views but lacks open space for tourists to step on and tour. Reducing the lawn area has little effect on the ecological diversity but can greatly increase the sightseeing space. Although the existing studies provide solid support for quantitative analysis of ecological and social benefits of urban parks, the efficient use of Taihu Park to provide ecological and social benefits still requires further exploration.

## Data Availability

The data presented in this study are available on request from the corresponding author.

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
