# Peer review of "Research on the Development of Urban Parks Based on the Perception of Tourists: A Case Study of Taihu Park in Beijing"

_ijerph, 2022, doi:10.3390/ijerph19095287_

Round 1

Reviewer 1 Report

Dear Authors,

The paper is interesting, but fig. 1 and 2, could not be as in actual version. You have to marked Yours country on Word map, and later add maps in more local scale. The map on fig. 2 show nothing - it is not visible. Please add also scale and sign of north directions.

I see hypothesis - it is really good for paper, but You have to use them in discussion chapter. You have to discisse if Yours hypothesis were good formulated or not etc...

Author Response

RC: Reviewer's comment, AR: Authors' response

RC: The paper is interesting, but fig. 1 and 2, could not be as in actual version. You have to marked Yours country on Word map, and later add maps in more local scale. The map on fig. 2 show nothing - it is not visible. Please add also scale and sign of north directions.

AR: We thank the reviewer for the valuable comments.

We've changed the version of the map and added a map with China labeled,all maps have add local scale and sign of north directions.And we have changed fig.2 and added another map to illustrate the location of Taihu Park.

RC: I see hypothesis - it is really good for paper, but You have to use them in discussion chapter. You have to discisse if Yours hypothesis were good formulated or not etc...

AR: Based on your comments, we have added the Discussion section to tell the hypothesises are good formulated.

Changed part in Manuscript:
The results support the hypothesis 1 to hypthesis 5, accessibility, environment perception, sevirce and quality perception and tourist loyalty have postive impact on tourist satisfaction, and tourist complain has a negative impact on tourist satisfaction. (Page15, paragraph4)

Reviewer 2 Report

There is originality in the investigated paper as there is less research report about management and sustainable development of urban parks as important tourist places. Literature review are relatively rich and updated. 

Methodology: Hypothesis 3: Tourist loyalty has a significant positive impact on tourist satisfaction; 177. What does really mean “tourist loyalty”. It should be explained.

209 questionnaires seems to be a small research figure, considering the fact that the parks are visited by thousands of tourists. It should be An estimate of the number of tourists visiting the park over a period of time should be given. What estimated % of all visitors to the park in April 2021 were respondents? It should be explained on what basis the number of respondents was determined and considered as sufficient.

The final conclusions are a bit too general. Conclusions should refer to the hypotheses made based on the obtained figures. It is recommended to continue research in order to improve the management of the development of parks, based on a much larger number of respondents. The presented research is a good basis for starting the implementation of effective management of the development of parks, taking into account the real needs of tourists and the creation of an effective model of infrastructure development.

Author Response

RC: Reviewer's comment, AR: Authors' response

RC: There is originality in the investigated paper as there is less research report about management and sustainable development of urban parks as important tourist places. Literature review are relatively rich and updated. 

AR: We thank the reviewer for the valuable comments.

RC: Methodology: Hypothesis 3: Tourist loyalty has a significant positive impact on tourist satisfaction; 177. What does really mean “tourist loyalty”. It should be explained.

AR: We have explained tourist loyalty in more detail in Research Hypothesis (2.2).

RC: 209 questionnaires seems to be a small research figure, considering the fact that the parks are visited by thousands of tourists. It should be An estimate of the number of tourists visiting the park over a period of time should be given. What estimated % of all visitors to the park in April 2021 were respondents? It should be explained on what basis the number of respondents was determined and considered as sufficient.

AR: We have added  a discussion of the proportion of respondents and their rationale to 2.2.

Changed part in Manuscript:
According to the data provided by the management personnel of Taihu Park, there were about 6000 visitors in April 2021, with an average of 200 visitors per day. We collected 33 questionnaires every day, with a sampling rate of about 16.9%, which is enough com-paring with other sampling tests (7.6% and 17.2%) [55,56]. (Page7, paragraph2)

RC: The final conclusions are a bit too general. Conclusions should refer to the hypotheses made based on the obtained figures. It is recommended to continue research in order to improve the management of the development of parks, based on a much larger number of respondents. 

AR: We have rewritten the conclusions and refined the content on the basis of the obtained figures. We are sorry that the survey was conducted last year and no more data can be collected at this time.

Changed part in Manuscript:
Taking Taihu Park as an example, this paper analyzes how tourists' perceptions affect the overall satisfaction of tourists through the data collected by questionnaires and the method of structural equation model. , and puts forward constructive suggestions for the management and development of Taihu Park based on the results. It arrived at the following conclusions:
(1)    Tourist satisfaction plays an important decisive role in the sustainable development of the park. In order to improve the urban status of the park, it is necessary to improve tourist satisfaction.
(2)    Environmental perception, service and quality perception, accessibility, and tourist loyalty all have a significantly positive impact on tourist satisfaction. Therefore, to improve the tourist satisfaction, environmental perception, service and quality perception, accessibility, and tourist loyalty must be enhanced.
(3)    Tourist complaints is in the opposite direction as the change of tourist satisfaction,whose rise will reduce tourist satisfaction.
According to the research results of the tourist satisfaction and its influencing factors, the following suggestions are made to enhance the quality of Taihu Park :
(1)    At the park’s manager level, Taihu Park can add self-service vending machines in the park to improve service and quality perception, increase the accessibility by increasing the number of parking spaces and improve its software and hardware service quality to reduce tourists' complaints.
(2)    From the government’s perspective, it is still up to the government to make efforts to enhance the status of the urban parks. Therefore, the government should rely on its own influence to increase publicity for Taihu Park, expand its influence, attract foreign investment, and form a development model in which the market promote the park. In addition, the government should adopt the necessary financial means to encourage Taihu Park maintain the trees and instructures greatly , improve the tourist satisfaction by giving more financial support and imroving the standard of management subsidy. (Page16, paragraph3)

RC: The presented research is a good basis for starting the implementation of effective management of the development of parks, taking into account the real needs of tourists and the creation of an effective model of infrastructure development.

AR: We thank the reviewer for the valuable comments.

Reviewer 3 Report

It is not clear to me how the authors define "tourists"?It should be clarified. If the authors have included in the study not only tourists but also residents (and this is how I understand it), the title and relevant content of the article should be remodeled.

The discussion should be deepened - Is it possible, according to the authors, and in the light of other studies, to simultaneously develop parks as recreation places and maintain their ecological functions? Where are the possible synergies, and where are the trade-offs?

And also, the editorial comment: Figure 2 in its present form is not very informative; at least a scale should be added. 

Author Response

RC: Reviewer's comment, AR:Authors' response

RC: It is not clear to me how the authors define "tourists"?It should be clarified. If the authors have included in the study not only tourists but also residents (and this is how I understand it), the title and relevant content of the article should be remodeled.

AR: We have clarified the "tourists" in the "Data Collection" (2.3).

Changed part in Manuscript:
To ensure the consistency of the samples, the respondents were restricted to tourists who had just finished visiting Taihu Park. Whether residents nearby or tourists far away, they are regarded as tourists as long as they enter the park, because the park is the subject we research, the difference of people's addresses is not a classification standard. (Page7, paragraph2)

RC: The discussion should be deepened - Is it possible, according to the authors, and in the light of other studies, to simultaneously develop parks as recreation places and maintain their ecological functions? Where are the possible synergies, and where are the trade-offs?

AR: We have added the discussion about it in "Conclusions and Suggestions" according to others' researches. 

Changed part in Manuscript:
Urban parks have two essential functions: biodiversity conservation and recreation provision, which have attracted increasing attention in spatial allocation of urban green space and city planning. Biodiversity conservation of urban parks may significantly beautify the environment, improve the recreation experience and health well-being of urban residents [65] while resident recreation behavior will impose a certain disturbance on the protection of urban green habitats [66]. A large number of studies have measured the function of urban parks through selecting the ecological connectivity and spatial accessibility, which provide a scientific basis for formulating multi-functional protection strategies for urban parks. And try to identify tradeoff relationship of urban parks in terms of ecological and social functions in Patero ranking method [67,68,69]. For Taihu Park, it is mainly positioned as an urban park to meet the leisure and exercise needs of nearby residents because of considering spatial accessibility, but it also maintain ecological function to beautify the environment to provide better experience. However, it should abandon some ecological functions ,such as reducing lawn area to provide tent building area, to ensure the carrying capacity of tourists for its position. Although the existing studies provide solid support for quantitative analysis of ecological and social benefits of urban parks, the tfficient use of Taihu Park to provide ecological and social benefits still require further exploration. (Page16, paragraph5)

RC: And also, the editorial comment: Figure 2 in its present form is not very informative; at least a scale should be added. 

AR: We have changed the Figure 2 and added scale and signs of north.

Reviewer 4 Report

  1. Since this is a "research" article, it is quite important to demonstrate the argument and critique capacity of academic quality, thus using any table like Table 1, that lists prior research is not be appropriate.  
  2. The lack of literature review with regard to all the research variables shies away from the value of this study.
  3. It is not clear how can any respondent respond to the questions.  Was it a Likert 5-point scale? 7-point? or 10-point?  from "strongly agree" to "strongly disagree" or from "very satisfied" to "very unsatisfied" ?  
  4. Since the sampling was conducted in a random fashion, it is not clear how the random process was conducted.
  5. Similar topics of research can be found and traced from prior publications, it is very important for the authors to compare and contrast this study as opposed to the prior ones?
  6. Some wordings may not be up to academic writing.  For instance, I suppose the word "true" in Table 9 should be "supported". 

Round 2

Reviewer 3 Report

The authors incorporated my comments into the revised text.

I think that further research should cover the relationship between various functions of city parks (synergies, trade-offs, bundles).

Author Response

This manuscript is a resubmission of an earlier submission. The following is a list of the peer review reports and author responses from that submission.

Round 1

Reviewer 1 Report

The article entitled "Research on the Development of Urban Parks Based on the Tourist Perception: A Case Study of Beijing Taihu Park" presents a good research proposal and is very relevant to the field of urban environment and health promotion.

The introduction of the work is clear and presents relevant theoretical foundation, although it could be improved. The statistical method and sampling effort is interesting, however, the authors were unable to adequately express and discuss the results. The text is confused in many parts and must be rewritten, including more related to the hypothesis test raised.

The conclusion also presents consubstantial problems.

I suggest that the authors rewrite the work and then present it to the journal.

Reviewer 2 Report

Dear Authors,

The paper "Research on the Development of Urban Parks Based on the Tourist Perception: A Case Study of Beijing Taihu Park" is interesting, but it's need some improvments. The chapter "Discussion" need huge modifications. You have to discussed yours results and compare them with others publications.

The others smaller gaps:

  • You should to add map with localization of the park in global and local scale
  • in the literature will be good to add these papers: 

    Mandziuk A. et al. Social Preferences of Young Adults Regarding Urban Forest Recreation Management in Warsaw, Poland Forests 2021, 12(11), 1524; https://doi.org/10.3390/f12111524

    Mandziuk et al. The willingness of inhabitants in medium-sized city and the city’s surroundings settlements to pay for recreation in urban forests in Poland, iForest-Biogeosciences and Forestry, 2021, vol. 14, nr 5, s.483-489. DOI:10.3832/ifor3758-014